# The Effect of the COVID-19 Pandemic Movement Restrictions on Self-Reported Physical Activity and Health in New Zealand: A Cross-Sectional Survey

**DOI:** 10.3390/ijerph18041719

**Published:** 2021-02-10

**Authors:** Rebecca M. Meiring, Silmara Gusso, Eloise McCullough, Lynley Bradnam

**Affiliations:** 1Department of Exercise Sciences, Faculty of Science, University of Auckland, Auckland 1023, New Zealand; s.gusso@auckland.ac.nz (S.G.); emcc157@aucklanduni.ac.nz (E.M.); lynley.bradnam@auckland.ac.nz (L.B.); 2Movement Physiology Research Laboratory, School of Physiology, University of the Witwatersrand, Johannesburg 2193, South Africa

**Keywords:** COVID-19, physical activity, movement restriction, anxiety, mental wellbeing, motivation

## Abstract

This study describes self-reported physical activity (PA), motivation to exercise, physical and mental health and feelings towards PA during the March-May 2020 COVID-19 lockdown in New Zealand. Adults over the age of 18 years (n = 238; 80.2% female) completed the International Physical Activity Questionnaire (IPAQ), the Behavioural Regulation in Exercise Questionnaire 3, the Short Form-36 and open-ended questions about PA through an anonymous online survey. Regular exercise was undertaken by 85% of respondents prior to lockdown, but only 49.8% were able to maintain their usual level of PA. Although respondents were considered sufficiently physically active from the IPAQ, 51.5% reported not being able to maintain their usual level of PA primarily due to the closure of their gym facilities. Sixty percent of respondents reported that PA had a positive effect on their overall wellbeing. When asked to specify which aspects of wellbeing were affected, the effect on mental health was reported the most while the effect on body image or fitness was reported the least. Strategies to increase or maintain engagement in physical activity during lockdowns should be encouraged to promote positive mental health during the COVID-19 pandemic.

## 1. Introduction

Substantial evidence exists to support the benefit of regular exercise and physical activity (PA) on all aspects of physical and mental health [1,2]. The current physical activity guidelines recommend participation in a minimum of 150 min of moderate to vigorous physical activity (MVPA) per week for healthy adults, older adults and those living with chronic disease [3]. The benefits for adults to engage in this amount of physical activity include reductions in the risks of all-cause mortality, cardiovascular disease, type 2 diabetes, dementia and cognitive dysfunction, reductions in anxiety symptoms, as well as improvements in quality of life and physical function [3]. There is however a high prevalence of physical inactivity globally with approximately only 50% of healthy adults worldwide meeting these recommendations [4]. Lack of time [5], family responsibilities [6], fatigue or feelings of weakness, lack of motivation, confidence, self-discipline and unaffordable exercise programmes as well as not making exercise a priority and health issues are well-documented barriers to sufficient PA participation in healthy adult populations [6,7,8].

Coronavirus disease (COVID-19) is an infectious disease caused by a newly discovered coronavirus [9]. The Director General of the World Health Organisation declared COVID-19 a global pandemic on 11 March 2020, which resulted in significant movement restrictions globally and within nations, and this crisis has impacted adults’ ability to engage in the recommended levels of PA and exercise. Some international surveys were conducted early on during the COVID-19 pandemic to investigate the effect of the lockdowns on self-reported PA levels [10,11,12,13,14,15,16,17,18,19,20,21,22,23,24,25,26]. These studies have indicated significant declines in self-reported physical activity levels during COVID-19 restrictions compared to before. Other studies have reported on the effects of the overall pandemic on PA levels in the context of mental health, specifically anxiety and depressive symptoms [27,28,29,30]. Studies that have investigated the effect of the overall pandemic on PA and well-being, found that reductions in PA and increases in sedentary behaviour (SB) were associated with poorer mental health and increased anxiety [14,16,18,21,24,31].

On the 25 March 2020, New Zealand moved into their highest COVID-19 alert level [32]. People were instructed to stay at home unless activities were deemed essential however, safe recreational activity in local areas was allowed if social and physical distancing rules were adhered to. In fact, one of the New Zealand government’s clear recommendations when the country entered into a state of restricted activity on 26 March, was for the general population to exercise safely in the vicinity of their homes [32]. There was, however, no direction given on the amount of physical activity and exercise to engage in to receive the positive benefits to physical and mental health. New Zealand moved to alert level 3 on 27 April 2020, which still imposed the same restrictions as alert level 4, except visits from close relatives or health care workers were allowed. The lockdown was lifted to pre-pandemic lifestyles but with border closure on 13 May 2020, after a total of 50 days.

The levels and underlying reasons for those levels of exercise or physical activity during the March-May COVID-19 lockdown of 2020 in New Zealand are unknown. The aim of this study was to describe the self-reported physical activity levels, motivation to exercise and physical and mental health of New Zealanders and to obtain qualitative information on the feelings towards physical activity and exercise during lockdown.

## 2. Materials and Methods

This was a cross-sectional study design in which an anonymous survey was distributed to adults over the age of 18 years in the general adult NZ population. This age group and population was chosen as no research has yet been undertaken regarding the impact of COVID-19 on health outcomes in New Zealand.

### 2.1. Survey Participants and Distribution

Invitation messages were posted on open community and neighbourhood social media, as well as on online community notices from April 2020 until June 2020. Participants were invited to access the survey information and consent was assumed if the participant completed and submitted the survey. The survey (Qualtrics XM, Qualtrics LLC) consisted of demographic questions of the participant as well as questions on the individual’s living situation under the strictest lockdown levels during the COVID-19 pandemic restrictions in New Zealand. Participants also completed a series of questionnaires that collected information on self-reported physical activity, motivation to exercise and quality of life. Responses were recorded anonymously. The study was approved by the University of Auckland Human Participants Research Ethics Committee (approval number: 024628).

### 2.2. Questionnaires

All questionnaires were modified to refer to the respondents’ experience during the 7-week lockdown. A modified form of the International Physical Activity Questionnaire (IPAQ) was used to collect information on domain-specific physical activity and sedentary behaviour over the last seven days [33]. The modified questionnaire requested that respondents specifically refer to a seven-day period during lockdown alert levels 3 or 4 which were the highest lockdown alert levels in New Zealand. Time reported walking and spent in moderate and vigorous activity was calculated in minutes per day for an average week during lockdown.

The Behavioural Regulation in Exercise Questionnaire (BREQ-3) is based on self-determination theory [34] and was used to assess the intrinsic and extrinsic motivation to exercise during lockdown [35,36]. The BREQ-3 is a 24-item self-report measure which contains six styles of motivation and respondents are asked to rate each question on a 4-item Likert scale (1 = not true for me, 3 = sometimes true for me, 5 = very true for me). The six styles of motivation are amotivation, external regulation, introjected regulation, identified regulation, integrated regulation and intrinsic motivation. A mean score for each motivation style is then calculated.

The short-form 36 (SF-36) was used to measure overall quality of life during lockdown across eight domains including physical functioning, bodily pain, role limitations due to physical health problems, role limitations due to personal or emotional problems, emotional well-being, social functioning, energy/fatigue and general health perceptions [37,38]. The questionnaire was scored using the RAND-36 protocol and a summary component score for physical health and mental health was calculated out of 100 [38]. A higher score indicates a more favourable health state [38].

Finally, open-ended questions were also included in the survey to obtain demographic information such as age, gender, ethnicity and work status, and qualitative data on respondents’ feelings towards their physical activity engagement or lack thereof during lockdown. These questions asked whether participants believed their activity levels had changed during compared to before the imposed lockdown and asked for reasons why that may or may not have been the case.

### 2.3. Data Reduction and Analysis

Data were summarised and are presented descriptively. Analysis of the main outcome variables of physical activity, motivation and quality of life was conducted using age group, living arrangement, sex and whether the individual had carer responsibility for children under the age of 18 years as factors. A one-way ANOVA with Bonferroni post-hoc analysis was used to analyse three or more groups (age and living arrangement groups). A Kruskal-Wallis test with Dunn’s post-hoc test was used to determine group effects for three or more groups for motivation and quality of life scores as a non-parametric test. An unpaired t-test was used to analyse the difference between two groups (between sexes or between those having to or not having to care for children under the age of 18). For non-parametric data, a Mann-Whitney U test was used to compare differences between 2 group (sex and caring for children <18 years). Finally, the answers to open-ended questions were analysed descriptively. Data were analysed using StataIC (version 15.1, StataCorp, TX, USA).

## 3. Results

### 3.1. Demographics

Of the 263 respondents who started the survey, 248 completed (80.6% females) the survey. After scoring and cleaning the data obtained from the IPAQ according to standard protocols (www.ipaq.ki.se, accessed on 19 November 2020), we obtained 196 complete responses on self-reported physical activity levels. The demographics of the 196 survey respondents is shown on Table 1. Approximately 70% of the study participants were under the age of 50 and of European ethnicity. When asked about their living arrangements, 15.8% lived alone and the rest of the cohort shared their household with another individual (partner, flatmate or relative), 73.0% had children. 79.1% of respondents were working during lockdown.

### 3.2. Physical Activity

From the cohort (n = 195), respondents reported spending a mean (SD) time of 62.4 (43.0) min walking, 125.6 (64.9) min in moderate intensity exercise and 34.3 (43.0) min in vigorous intensity activity per day in an average week during lockdown. Table 2 shows the self-reported times spent walking and in moderate and vigorous physical activity for each classification of age, gender, living arrangement and children in the household.

Males reportedly spent approximately 16 min more time in vigorous physical activity per day during lockdown than females.

### 3.3. Motivation to Exercise

The scores from the six styles of motivation calculated from the BREQ-3 questionnaire are presented in Table 3. Overall, respondents reported higher scores (“very true for me”) in the questionnaire items that related to intrinsic motivation.

### 3.4. Quality of Life

Overall, the physical health score (SF-36) of respondents was 56.5 (50.1–61.5) and the mental health score was 38.5 (21.1–50.5) (median IQR) (Table 4).

### 3.5. Qualitative Data

Respondents were asked to answer questions related to their experience of physical activity and exercise during the lockdown period. We obtained complete data from 238 respondents. 81.9% of respondents rated their overall well-being during the COVID-19 lockdown period as being either excellent or good. Over half of the respondents (54.9%) reported that they had started a new exercise regime during lockdown.

Eighty-five percent of respondents reported that they took part in regular exercise or physical activity prior to lockdown. When these respondents were asked if they had been able to maintain their level of physical activity during lockdown, 48.5% said they were, while 51.5% said they were unable to maintain or increase their PA level during this period. The most reported reason given for being able to increase or maintain level of PA during lockdown, was more time was available from not having to commute into work. Other responses provided were that some people felt they were able to adapt their type of exercise to outdoor activity. Others felt that lockdown had motivated them to do some type of activity/made a conscious effort to get outside and do some physical activity or exercise. From those (51.5%) who reported not being able to maintain their PA during compared to before lockdown, the most reported reason given was that gym facilities were closed (Table 5). Other reasons included fear of going outside, no social aspect of engaging in PA or exercise for them, lack of motivation, not enough equipment at home, or people had a work commute that could not be replaced by other activity during lockdown.

Of the respondents who reported regularly taking part in exercise or physical activity, 90.1% reported that their engagement in PA and exercise affected their overall wellbeing during lockdown. These respondents qualified how their engagement in PA affected their overall wellbeing. The qualitative response provided was classified as PA having a positive effect on wellbeing, a negative effect on wellbeing, or no response was provided. Positive effects were responses such as, “it gave me a break from being inside all the time” or “it was good to get some time away from my family!”. While negative effects were responses such as, “No teams sports” or “Elite level swimmer with 20 + hrs of training a week normally but as soon as lockdown started we could no longer swim and was only a few weeks out from Olympic trials so a lot of hard work to not be able to show for it”. Positive effects of PA and exercise on well-being were reported by 57.6% of respondents, while 17.2% reported that their engagement with PA or lack thereof, during lockdown, had a negative effect on their wellbeing. 25% did not elaborate on their answer. Positive effects on wellbeing were further sub-classified into an impact on one of three domains: (1) body image or fitness, (2) mental wellbeing or (3) other. More than half of respondents (54.7%) who reported PA impacting positively on their wellbeing commented on their mental health or mental wellbeing.

Of the remaining 36 respondents who reported not engaging in regular PA or exercise prior to lockdown, half (50.0%) reported that regular PA had a positive impact on their wellbeing during lockdown mainly because they had started doing more physical activity such as walking during this time.

## 4. Discussion

The aim of the present study was to describe self-reported physical activity, motivation to exercise and physical and mental health of New Zealanders during the March-May 2020 COVID-19 lockdown. This study also aimed to explore some of the feelings towards physical activity during this lockdown period. Our study findings indicated that the majority of respondents took part in regular physical activity prior to the lockdown period, and that only half of those were able to maintain their current level of activity during the lockdown period. These individuals reported that the main barrier to PA was that their exercise facilities (e.g., gym) were closed, followed by being afraid to go out and being unable to exercise in a social group or with their friends. Previous studies have reported that a lack of time, cost, and a lack of knowing what to do are common barriers that prevent people from engaging in physical activity [7,39,40]. Some of these barriers to physical activity were exemplified in the change in environment attributable to the stay-at-home orders during the COVID-19 pandemic.

The reduction in PA reported in our study is similar to findings observed in other international studies during lockdowns or movement restrictions [17,18,19,20]. However, there have been some surveys that have highlighted increased engagement in physical activity despite the pandemic. One study conducted in Canada [41] that measured activity behaviours using wearable fitness devices, also found that all intensities of activity (number of steps, light and moderate to vigorous intensities) declined significantly immediately following the announcement of the pandemic. However, moderate to vigorous PA returned to pre-pandemic levels six weeks after the start of the pandemic, while light activity and the number of steps remained lower. A similar study in Greece also found a levelling off of the decline in PA towards the end of lockdown [22]. Furthermore, two other studies showed a discrepancy in activity change for some people during the COVID-19 lockdowns. In an online survey conducted in Canada, results indicated that 40.5% of individuals who were inactive before lockdown became less active than they had been previously. However only 22.4% of previously active individuals became less active following lockdown. Furthermore, the authors found that a significant proportion of both previously inactive and active individuals became more active during lockdown [10]. In another study that collected 13,515 responses to a survey in Belgium [11], a general increase in exercise frequency, but also sedentary behaviour was observed during lockdown. Adults under the age of 55 years old who were considered low active before lockdown self-reported to exercise more during the lockdown. Among people who were already active before the pandemic, only those who were older than 55 years, had low education, engaged in social exercise, or those who were not using online tools to exercise, self-reported to exercise less during the lockdown [11]. Similarly, from some of the qualitative responses provided in our survey, certain people recognized the need to remain or become more physically active during lockdown.

Respondents in our survey would be considered as being sufficiently physically active according to the current World Health Organisation (WHO) physical activity guidelines for adults [42]. In a similar survey conducted in four high-income countries including New Zealand, it was found that adults in New Zealand had the least change to their physical activity in the early stages of the lockdown [43]. In New Zealand the movement restrictions still allowed for engagement in exercise in the vicinity of people’s homes therefore this may have also contributed to allowing people to continue with some physical activity during lockdown. In a survey of young adults in Italy, maintaining adequate levels of PA was associated with prior undertaking of an active lifestyle [19]. People who qualitatively provided reasons regarding their willingness to engage in PA even though their regular exercise option was not available, together with the average level of PA that was reported in our survey suggests that respondents were already motivated to engage in exercise. Indeed the results from the BREQ-3 questionnaire, indicated that respondents were intrinsically motivated to exercise suggesting that participants were more likely to engage in physical activity of their own accord [26].

The COVID-19 lockdown has also impacted people’s mental health worldwide [14,16,18]. In our survey, those respondents who continued to maintain their usual level of physical activity throughout the lockdown, recognized the importance of PA on their well-being especially mental health. This is in line with other research that has shown the benefits of physical activity on mental health [18,43,44,45] and in particular with a survey conducted in New Zealand that showed positive correlations between self-reported PA and well-being [43]. In this study, people who had negative or no change in their PA from pre-during lockdown, reported worse well-being, depression, stress and anxiety compared to those who had positive changes in PA [43]. During lockdown in Canada, men and women who were surveyed, reported very good or excellent mental and physical health if they were exercising outdoors and if they reduced their screen time [14]. A study conducted in Japan during lockdown showed that a reduction in physical activity following COVID-19 restrictions was associated with an increased risk of a lower mental component health-related quality of life score in elderly people in Japan [30]. A similar association was seen in adults in Australia, who reported that overall there was a decrease in physical activity levels which was associated with higher depression, anxiety and stress symptoms [27].

The qualitative analysis of the impact of lockdown on mental health was reflected quantitatively in the mental health scores from the SF-36 reported in our study. The scores indicated that respondents had a poor self-perception of mental health during lockdown while they reported an average self-perception of physical health. This finding aligns with several other studies that have indicated increased symptoms of anxiety and depression, and reduced perception of quality of life during nation-wide COVID-19 lockdowns in adults and older adults [14,16,21,31]. However, the mental health scores in our respondents were considerably lower compared to a study that assessed health related quality of life using the SF-8 in online survey in China [21]. This finding is surprising however may not be associated with ability to engage in physical activity but may rather relate to the stricter rules regarding family and friend visits or social gatherings.

### Limitations

Due to the nature of this study being an online survey, selection biases are a limitation as the participants who volunteered to complete this survey may not be an accurate representation of the New Zealand population. A lack of ethnic diversity also suggests limitations in sampling, as the Auckland region has higher proportions of Asian, Māori, and pacific ethnicities, which were not proportionately represented in this study. Self-reported measures are also a limitation of the present study due to the tendency for participants who are already sufficiently physically active to respond to a survey regarding physical activity. A third limitation of the study was the reliance on participants to recalling previous activities, such as physical activity levels prior to lockdown.

## 5. Conclusions

Overall, the movement restrictions put in place in New Zealand due to the global COVID-19 pandemic impacted the level of PA which may then have had effects on physical and mental health. A contributing factor to the ability to maintain engagement in PA could also have been the type of messaging that was delivered whereby by people were actively encouraged to engage in some exercise but within a certain distance of their home. This may explain why most respondents in the survey rated their overall wellbeing as excellent or good. However, there was a certain proportion of respondents who were unable to engage in activity due to closure of exercise and sporting facilities. Further questions remain as to whether there may be lingering anxiety in people wishing to take up exercise especially those who fall within the vulnerable category and what considerations may be required for additional support for people who may continue to not engage in physical activity during the lockdown period of the COVID-19 pandemic.

## Figures and Tables

**Table 1 ijerph-18-01719-t001:** Respondent demographics.

Demographic	N (%)
Participants with complete survey data	196
Age range	
18–29	54 (27.6)
30–39	44 (22.5)
40–49	33 (16.8)
50–59	32 (16.3)
60–69	23 (11.7)
70+	10 (5.1)
Gender identity	
Female	158 (80.6)
Male	37 (18.9)
Gender diverse	1 (0.5)
Ethnicity	
Māori	4 (2.0)
European	145 (74.0)
Asian	14 (7.1)
Pacific peoples	3 (1.5)
Middle Eastern/Latin American/African	5 (2.0)
Other	26 (13.3)
Living arrangement	
Alone	31 (15.8)
With a partner	55 (28.0)
With a partner and children	52 (26.5)
With children	8 (4.1)
With flatmates/house-share	18 (9.2)
With family and close relatives	28 (14.3)
Other	4 (2.0)
Children	
Yes	143 (73.0)
No	53 (27.0)
Working *	
Yes	72 (79.1)
No	19 (20.1)

* 105 people did not respond to this question.

**Table 2 ijerph-18-01719-t002:** Self-reported physical activity.

Demographic	Walking (Min/Day)	Moderate (Min/Day)	Vigorous (Min/Day)
Age range			
18–29	59.0 (42.8)	108.7 (67.8)	39.2 (41.6)
30–39	56.6 (41.5)	138.2 (58.6)	30.0 (31.2)
40–49	62.3 (39.8)	116.1 (67.4)	25.0 (29.0)
50–59	58.9 (36.2)	121.1 (68.5)	34.2 (51.7)
60–69	73.3 (54.0)	151.3 (51.0)	39.1 (49.9)
70+	90.0 (49.0)	147.0 (64.0)	48.0 (77.7)
Gender *			
Female	62.4 (41.0)	127.8 (63.2)	31.2 (39.7)
Male	62.4 (51.6)	116.5 (72.1)	47.8 (53.3)
Living arrangement			
Alone	57.2 (58.4)	118.9 (70.0)	34.1 (55.8)
With a partner	58.2 (41.1)	111.2 (75.9)	31.1 (37.7)
With a partner and children	50.2 (43.0)	103.0 (78.5)	19.5 (31.6)
With children	53.3 (45.6)	103.3 (74.8)	28.3 (50.0)
With flatmates/house-share	36.0 (51.9)	52.3 (69.0)	25.3 (44.7)
With family and close relatives	37.4 (36.8)	91.5 (78.5)	25.5 (34.9)
Other	35.8 (44.8)	100.8 (89.9)	36.7 (46.8)
Children			
Yes	67.4 (44.3)	132.1 (65.1)	33.7 (48.6)
No	58.0 (41.7)	120.0 (64.5)	34.9 (37.6)
Working **			
Yes	67.7 (45.8)	126.8 (68.0)	35.9 (50.0)
No	66.2 (39.1)	152.1 (48.7)	25.3 (42.7)
No response	58.0 (41.5)	120.5 (64.5)	34.9 (37.4)

Data are mean (SD). * 1 respondent identified as gender diverse. ** 105 people did not respond to this question.

**Table 3 ijerph-18-01719-t003:** BREQ-3 scores.

	Amotivation	External Regulation	Introjected Regulation	Identified Regulation	Integrated Regulation	Intrinsic Regulation
Total	1.0 (1.0–1.3)	1.3 (1.0–2.0)	3 (2.3–3.8)	4.5 (3.8–4.8)	3.6 (2.6–4.5)	4.0 (3.3–4.8)
Age Group						
18–29	1.0 (1.0–1.3)	1.5 (1.0–2.0)	3.3 (2.3–4.0)	4.5 (4.0–5.0)	3.9 (3.3–5.0)	4.3 (3.8–5.0)
30–39	1.0 (1.0–1.3)	1.3 (1.0–2.0)	3.3 (2.5–4.0)	4.3 (3.6–4.9)	3.3 (2.3–4.6)	3.8 (3.0–4.8)
40–49	1.0 (1.0–1.0)	1.3 (1.0–1.6)	2.8 (2.3–3.5)	4.3 (3.5–4.8)	3.5 (2.5–4.0)	3.5 (3.0–4.0)
50–59	1.0 (1.0–1.0)	1.3 (1.0–2.0)	2.8 (1.6–3.8)	4.5 (3.5–4.8)	4.0 (2.6–4.5)	4.0 (3.4–4.6)
60–69	1.0 (1.0–1.0)	1.3 (1.0–1.7)	3.0 (2.3–3.5)	4.8 (4.0–5.0)	4.3 (3.3–5.0)	4.0 (3.8–5.0)
70+	1.0 (1.0–1.0)	1.1 (1.0–1.5)	3.4 (2.8–4.5)	4.5 (4.0–5.0)	3.8 (3.0–5.0)	3.8 (3.0–4.5)
Gender *						
Female	1.0 (1.0–1.0)	1.3 (1.0–2.0)	3.0 (2.3–3.8)	4.5 (3.8–4.8)	3.5 (2.5–4.5)	4.0 (3.3–4.8)
Male	1.0 (1.0–1.3)	1.3 (1.0–1.5)	3.0 (2.3–3.5)	4.5 (3.8–4.8)	4.0 (3.0–5.0)	4.0 (3.5–4.8)
Living Arrangements						
Alone	1.0 (1.0–1.0)	1.0 (1.0–1.3)	2.8 (2.0–3.8)	4.5 (3.5–4.8)	4.0 (3.0–4.8)	4.0 (3.5–5.0)
With partner	1.0 (1.0–1.3)	1.5 (1.0–2.3)	3.0 (2.3–3.8)	4. 5 (3.8–4.8)	3.5 (2.3–4.0)	4.0 (3.0–4.8)
Partner & children	1.0 (1.0–1.3)	1.3 (1.0–1.9)	3.0 (2.3–3.6)	4.3 (3.4–4.8)	3.3 (2.3–4.0)	3.5 (3.1–4.3)
With children	1.0 (1.0–1.1)	1.0 (1.0–1.3)	3.3 (2.3–4.0)	4.5 (3.5–5.0)	4.1 (2.9–4.9)	4.1 (3.1–4.5)
Flatmates & houseshare	1.0 (1.0–1.3)	1.5 (1.3–2.0)	3.4 (2.8–3.8)	4.6 (4.0–4.8)	3.9 (3.5–5.0)	4.6 (4.0–5.0)
Family & close relatives	1.0 (1.0–1.0)	1.5 (1.0–1.5)	3.5 (2.3–4.5)	4.8 (3.8–5.0)	4.0 (3.4–5.0)	4.4 (3.5–5.0)
Other	1.0 (1.0–1.1)	1.9 (1.5–2.5)	4.5 (3.4–4.5)	4.9 (4.6–5.0)	4.8 (4.5–5.0)	4.4 (3.9–4.9)
Children under 18						
Yes	1.0 (1.0–1.0)	1.3 (1.0–2.0)	3.0 (2.3–4.0)	4.5 (4.0–4.8)	4.0 (3.0–4.8)	4.0 (3.0–4.8)
No	1.0 (1.0–1.3)	1.3 (1.0–2.0)	3.0 (2.3–3.5)	4.0 (3.5–4.5)	3.0 (2.3–4.0)	3.0 (2.3–4.0)
Working **						
Yes	1.0 (1.0–1.0)	1.3 (1.0–1.6)	3.0 (2.3–3.8)	4.5 (3.5–4.8)	3.4 (2.5–4.4)	3.8 (3.3–4.4)
No	1.0 (1.0–1.0)	1.3 (1.0–2.0)	3.0 (2.0–4.0)	4.5 (3.5–4.8)	3.7 (2.5–4.8)	4.0 (3.0–4.8)
No response	1.0 (1.0–1.3)	1.5 (1.0–2.0)	3.0 (2.3–4.0)	4.5 (3.8–4.8)	3.8 (2.8–4.8)	4.0 (3.5–4.8)

Data are median (IQR). * 1 respondent identified as gender diverse. ** 105 people did not respond to this question.

**Table 4 ijerph-18-01719-t004:** Quality of life physical and mental health scores as measured using the SF-36.

	Physical Health Score	Mental Health Score
Age range (years)		
18–29	58.7 (51.0–63.0)	29.0 (8.8–43.7)
30–39	56.9 (48.9–62.2)	36.3 (17.5–49.4)
40–49	56.4 (49.3–60.7)	38.0 (26.1–45.5)
50–59	57.8 (54.7–61.0)	42.7 (22.5–52.9)
60–69	52.8 (45.9–60.0)	47.0 (33.6–52.8)
70+	51.9 (44.4–56.2)	49.5 (29.2–56.1)
Gender *		
Female	56.6 (50.4–61.6)	37.0 (20.7–50.6)
Male	56.2 (48.7–60.7)	42.2 (28.2–49.8)
Living arrangements		
Alone	54.3 (46.2–57.9)	47.1 (30.8–53.0)
With a partner	59.2 (52.0–62.3)	40.2 (20.6–51.4)
With a partner and children	56.3 (49.0–60.9)	37.9 (24.9–50.2)
With children	55.9 (48.8–56.9)	32.2 (11.1–48.4)
With flatmates/houseshare	60.1 (50.3–64.4)	23.1 (11.9–43.0)
With family and close relatives	55.9 (51.2–61.8)	35.5 (10.8–43.9)
Other	58.4 (52.5–63.6)	21.8 (14.6–37.4)
Children		
With	56.4 (48.7–61.1)	35.3 (21.7–49.6)
Without	56.5 (50.4–61.6)	39.1 (20.7–50.9)
Working **		
Yes	56.5 (50.6–60.3)	43.9 (27.4–52.9)
No	50.4 (44.3–56.2)	29.2 (24.2–49.8)
No response	58.2 (51.0–62.3)	36.8 (14.2–47.0)

Data are median (IQR). * 1 respondent identified as gender diverse. ** 105 people did not respond to this question. Physical and mental health scores are out of 100. A higher health score indicates a more favourable health state.

**Table 5 ijerph-18-01719-t005:** Reasons for not engaging in physical activity or exercise.

	%
My gym is closed	28.4
Nothing prevents me from engaging in exercise	20.3
I choose not to exercise	16.8
I am afraid to go out	10.8
I can’t exercise without my friends/social group	9.5
Lack of motivation	6.2
I do not know what to do	5.1
I do not have time	3.0

## Data Availability

The data presented in this study are available on request from the corresponding author. The data are not publicly available due to privacy reasons.

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
