# Peer review of "The Effect of the COVID-19 Pandemic Movement Restrictions on Self-Reported Physical Activity and Health in New Zealand: A Cross-Sectional Survey"

_ijerph, 2021, doi:10.3390/ijerph18041719_

Round 1

Reviewer 1 Report

The study investigated physical activity and mental health levels during the 7-week lockdown in NZ using an online survey. Major comment: It is not clear if the aim of the study is to compare PA and mental health during lockdown with pre-pandemic levels OR to describe differences in PA and mental health in different demographics during lockdown. Much of the results presented focusses on the later - i.e., it is not clear why the authors are statistically comparing different demographics (age, sex, living arrangements, etc). I would suggest simply presenting these data and not compare them as this does appear to be the aim of the study (based in Introduction and Discussion). Rather, the authors may consider presenting the qualitative data (with respect to pre-pandemic vs pandemic PA levels) according to demographic sub groups to identify why groups are most susceptible to the reduction in PA and mental health concerns. 

Other comments: 

  • what is meant by "PA having a positive impact on wellbeing"? This is stated throughout the manuscript but is not clear. Line 23-25, 
  • Line 26 in Abstract - suggest being more specific re how habitual PA was impacted. 
  • Line 92 - how were the questionairres modified? In what way?
  • Line 48 - comma typo
  • Line 50-51- the two sentences re safe exercise can be merged
  • Line 56-57 - not clear why authors are suggesting that the advice at the beginning of the pandemic should have focussed on amount and intensity of exercise? The word "safe" was presumably interpreted in context given the situation at the time?
  • Line 71 - SB?
  • Line 85 - please elaborate on information gathered re the "individuals situation"
  • Sample size - line 132 states 248 completed and were analysed but later line 142 states only 195 were included. Please explain all this in the one paragraph. Further to this, please elaborate on how data were cleaned according to standard protocols. Why were some data removed and how many for which parameters?
  • Line 136, change "while" to "and"
  • Table 1 - some numbers dont add up. e.g. children - 198+64 = 262 but state earlier that 248 completed survey. working -86+22+40 that did not respond = 148 only. Should include demographic information only for participants that were actually included in final cleaned analyses (N=195). 
  • is PA reported per day or per week? Time spent walking during the lockdown period appears to be very high - 1 hour per day. Is this standard in NZ? Did they survey ask for respondents to report on PA levels per day or per week? What is vigorous activity occurred only 1-2 times per week?  Also, Line 155 states 16 minutes more exercise per WEEK. Is this day or week?
  • Line 193 - sentence does not read well
  • Line 194-195 - is this compared with pre-pandemic levels or just in general?
  • Line 197: "Of those, 90.1% reported that their PA impacted on their overall well-being during lockdown." What does this mean?
  • Line 208: "18% reported a negative impact (or lack thereof)." Does this mean that 18% reported that lack of PA negatively impacted their wellbeing?
  • Line 215-217: not sure what this means. Please be specific about impacts (positive/negative?). "..63.9% (n=23) still reported that PA had some impact on their overall wellbeing during lockdown."
  • Line 223-225: comma typo and sentence overall need to be re-written.
  • Line 240-245: Combine with previous paragraph. 
  • Line 289: "all" adults?
  • Line 332: perhaps discuss levels of PA reported here in relation to other national/expected data. 
  • Line 346 - was information pertaining to underlying disease/"vulnerability" to COVID-19 captured in this study?

Author Response

Thank you to both reviewers for providing their valuable input. Please see our itemized responses to the feedback below.

The line numbers provided in the response refer to the line numbers in the manuscript with the track changes however there is also a manuscript with track changes for ease of reading.

Reviewer 1:

Thank you for your valuable input and suggestions to better clarify the manuscript.

The study investigated physical activity and mental health levels during the 7-week lockdown in NZ using an online survey. Major comment: It is not clear if the aim of the study is to compare PA and mental health during lockdown with pre-pandemic levels OR to describe differences in PA and mental health in different demographics during lockdown. Much of the results presented focusses on the later - i.e., it is not clear why the authors are statistically comparing different demographics (age, sex, living arrangements, etc). I would suggest simply presenting these data and not compare them as this does appear to be the aim of the study (based in Introduction and Discussion). Rather, the authors may consider presenting the qualitative data (with respect to pre-pandemic vs pandemic PA levels) according to demographic sub groups to identify why groups are most susceptible to the reduction in PA and mental health concerns. 

Reply: Thank you for raising this point. The aim of the study was to describe the differences in PA and mental health in adults in New Zealand during lockdown. We have amended the document to ensure our aim is clearly presented. We agree with the reviewer’s suggestion to describe the PA and mental outcomes rather than compare them. However, as mentioned by reviewer 2 it is appropriate to describe the qualitative data as the sample sizes between demographic groups are limited in number. We have therefore removed the comparison of the results between demographic groups and have presented the quantitative and qualitative data in the results section. Tables 1, 3 and 4 have been changed accordingly.

Other comments: 

  • what is meant by "PA having a positive impact on wellbeing"? This is stated throughout the manuscript but is not clear. Line 23-25, 

Reply: Thank you for raising this point. The positive and negative impact statements were responses to an open-ended question in which respondents were asked to explain whether and how they felt physical activity had impacted on their overall wellbeing during lockdown. We have changed positive impact to positive effect in the abstract on lines 25-26. We have also amended lines 44 to 47 in the introduction by adding examples of the documented positive effects that exercise and physical activity has on health and wellbeing. In lines 250-267 in the results section, we have more details on some of the responses that were classified either as a positive impact response or as a negative impact response.

  • Line 26 in Abstract - suggest being more specific re how habitual PA was impacted. 

Reply: Lines 21-25. We have added in the abstract that half of respondents who took part in reglar exercise or physical activity prior to lockdown were able to maintain or increase their engagement in PA.

  • Line 92 - how were the questionnaires modified? In what way?

Reply: Line 121-123: we have added the form that was modified to point respondents to answering the physical activity questions referring to a seven-day period during lockdown level 3 or 4, which were the highest COVID alert levels in New Zealand.

  • Line 48 - comma typo

Reply: The comma here has been deleted.

  • Line 50-51- the two sentences re safe exercise can be merged

Reply: Lines 70-73. These sentences have been merged and re-written as: “People were instructed to stay at home unless activities were deemed essential and for ‘safe’ recreational activity in local areas only if social and physical distancing rules were adhered to”.

  • Line 56-57 - not clear why authors are suggesting that the advice at the beginning of the pandemic should have focussed on amount and intensity of exercise? The word "safe" was presumably interpreted in context given the situation at the time?

Reply: We agree that the word ‘safe’ is mentioned in the context of minimising spread of the virus, and we have deleted the last part of the sentence in line79 - 81 “rather, than safety regarding exercise intensity or volume of activity”.

However given that the literature shows the benefits of physical activity and exercise to physical and mental health are associated with engaging in at least 150 minutes of moderate intensity activity, and given that several other studies have shown the negative effect of lockdowns on physical and mental health, we feel it is important to highlight the inadequacy of the public health messaging. We have added in lines 45-47, as per the suggestion of reviewer 2, a summary of the many benefits of adequate amounts of physical activity on physical and mental health. Furthermore we have moved the paragraph citing other literature on the impact of Covid-19 on physical activity levels earlier in the introduction to emphasise the reduction in activity that Covid-19 added.

  • Line 71 - SB?

Reply: Apologies for this oversight, we have defined sedentary behaviour (SB) in line 68.

  • Line 85 - please elaborate on information gathered re the "individuals situation"

Reply: Apologies for this oversight, we have added “living” here to refer to the individual’s living situation at the time of lockdown. Line 112.

  • Sample size - line 132 states 248 completed and were analysed but later line 142 states only 195 were included. Please explain all this in the one paragraph.

Reply: Thank you for this suggestion, we have written this in one section in lines 164-167. As suggested further we have only presented the demographics and mental health and motivation outcomes for the 196 complete responses (1 participant identified as gender diverse and we have not presented data on the one participant in the male/female sub-groups).

  • Further to this, please elaborate on how data were cleaned according to standard protocols. Why were some data removed and how many for which parameters?

Reply: Lines 163-166. We have added a reference to the scoring protocol that was used for cleaning and processing data obtained. The scoring protocol requires cases to be removed if the total number of days or minutes reported is missing, if the total number of days reported for all domains adds up to greater than 9, or if the sum total of all walking, moderate and vigorous time variables is greater than 960 minutes (16 hours). If any of those criteria were true, the case was excluded from analysis. The criteria assume that on average an individual of 8 hours per day is spent sleeping.

  • Line 136, change "while" to "and"

Reply: Line 169 changed “while” to “and”

  • Table 1 - some numbers dont add up. e.g. children - 198+64 = 262 but state earlier that 248 completed survey. working -86+22+40 that did not respond = 148 only. Should include demographic information only for participants that were actually included in final cleaned analyses (N=195). 

Reply: Thank you for picking this up and your suggestion. We have now only included the demographic, quality of life and motivation data for the 196 respondents who had complete physical activity data and Tables 1-4 have been changed to reflect this number. We believe that there are some valuable qualitative comments to be described from the total number of responses and therefore we have chosen to include the total number of qualitative responses in the results (n=238).

  • is PA reported per day or per week? Time spent walking during the lockdown period appears to be very high - 1 hour per day. Is this standard in NZ? Did they survey ask for respondents to report on PA levels per day or per week? What is vigorous activity occurred only 1-2 times per week? 

Reply: PA is reported per day. Respondents were asked to consider walking in all habitual domains of work, transport, domestic activity as well as leisure time activity, excluding moderate and vigorous intensity activity, therefore the total amount of time spent habitually walking (light intensity activity) in a day is not particularly high. The survey asked respondents to report on how many days walking took place during an average week in lockdown and then follows up by asking how much time is spent walking on one of those days. The total time spent walking is then summed for all the domains.

Also, Line 155 states 16 minutes more exercise per WEEK. Is this day or week?

Reply: Apologies for this oversight. This is per day during an average week in lockdown. We have corrected this in line 190.

  • Line 193 - sentence does not read well

Reply: Lines 226-227. We have split this sentence into 2 and corrected the typo in the second sentence.

  • Line 194-195 - is this compared with pre-pandemic levels or just in general?

Reply: Lines 228 - 229. This is not compared to pre-pandemic, it is only how people rated their overall well-being during the lockdown.

  • Line 197: "Of those, 90.1% reported that their PA impacted on their overall well-being during lockdown." What does this mean?

Reply: Apologies for not being clear with this statement. In lines 251-255, we have clarified this statement that 90.1% provided a qualitative response that described how their engagement in PA and exercise affected their overall wellbeing during lockdown.

  • Line 208: "18% reported a negative impact (or lack thereof)." Does this mean that 18% reported that lack of PA negatively impacted their wellbeing?

Reply: Again apologies for not making this clear. The reviewer is correct that the lack of PA in the respondents also negatively affected their wellbeing. This sentence is made clearer in lines 265-268.

  • Line 215-217: not sure what this means. Please be specific about impacts (positive/negative?). "..63.9% (n=23) still reported that PA had some impact on their overall wellbeing during lockdown."

Reply: We have clarified this as described above.

  • Line 223-225: comma typo and sentence overall need to be re-written.

Reply: Lines 234. Comma has been deleted and sentence rewritten.

  • Line 240-245: Combine with previous paragraph. 

Reply: Thank for pointing out this oversight. This sentence has been combined with the relevant paragraph and now appears on lines 242-246.

  • Line 289: "all" adults?

Reply: We understand that the use of all adults was not clear. We therefore amended our text in lines 346-348 accordingly.

“In our survey, when people were asked to report on their level of physical activity, we found that the survey participants would be considered as being physically active according to the current WHO physical activity guidelines for adults [44].”

  • Line 332: perhaps discuss levels of PA reported here in relation to other national/expected data. 

Reply: Thank you for this suggestion. We have discussed the levels of PA here in relation to another similar study and cited the relevant reference lines 348-351.

  • Line 346 - was information pertaining to underlying disease/"vulnerability" to COVID-19 captured in this study?

Reply: We understand the rationale for the reviewer’s inquiry however unfortunately our survey did not obtain information regarding underlying disease.

Reviewer 2:

Thank you for the opportunity to read this very interesting paper.  The paper has many strengths that should be of interest to the journal audience. However, it is not a significant sample although it has a significance of content and the methods and results have been relevance. 

Thank you for the opportunity to read this very interesting paper. The paper has much strength that should be of interest to the journal audience. Thus, the following suggestions are around enhancing the presentation for publication and clarifying aspects of the data and reporting. I will go by line number for the most part. If not, I will try to be as specific as possible in noting the area I am speaking about.

Abstract We strongly encourage authors to use the exactly number of words is required for the abstracts. Besides, it should be considered to include “motivation” as a keyword.

Reply: Thank you for your suggestion. We have shortened the abstract considering the new presentation of results as suggested by reviewer 1 and included “motivation” as a key word in line 36. Our abstract is now within the required word count.

I think you could add that not only you investigate the level of habitual physical activity but also to explore the reasons why they did or not.

Reply: Thank you for the suggestion. We have added in the abstract that we also aimed to explore the reasons for the PA levels and mental health during lockdown (Line 13).

  1. Introduction

[Line 40] Many studies have identified barriers and facilitator to exercise. Lack of time and so on….. Here, you could add some of the benefits that studies have mentioned.

Reply: Thank you for this suggestion. In lines 44-48, we have added information regarding the multiple benefits that physical activity has on physical and mental health. This added information also clarifies a comment made by reviewer 1.

You could compare or at least mention some examples of relevant studies about the relationship between: physical activity and mental health. I suggest some of these articles to read and write a brief comparison between before and after the COVID-19.

  1. Callow et al The Mental Health Benefits of Physical Activity in Older Adults Survive the COVID-19 Pandemic.
  2. Duncan et al Perceived change in physical activity levels and mental health during COVID-19: Findings among adult twin pairs.
  3. Faulkner et al Physical activity, mental health and well-being of adults during early COVID-19 containment strategies: A multi-country cross-sectional analysis.
  4. Jacob et al The relationship between physical activity and mental health in a sample of the UK public: A cross-sectional study during the implementation of COVID-19 social distancing measures.
  5. Pieh et al The effect of age, gender, income, work, and physical activity on mental health during coronavirus disease (COVID-19) lockdown in Austria.
  6. Schuch et al Associations of moderate to vigorous physical activity and sedentary behavior with depressive and anxiety symptoms in self-isolating people during the COVID-19 pandemic: A cross-sectional survey in Brazil.

Reply: We appreciate the time taken to provide us with these references. We have now included some of these studies into our manuscript accordingly (lines 57-68).

[Line 45] The Director of the World Health Organisation declared Covid-19 a global pandemic on 11 March 2020. It could be added just a few words about it such as: “The coronavirus disease (COVID-19) is a pathology considered a severe acute respiratory syndrome coronavirus-2 (SARS-CoV-2) induced by a new coronavirus”

Reply: Thank you for this suggestion. We have added this detail in line 55-56.

[Line 73-75] The aim of this study was to describe the self-reported physical activity levels, quality of life and motivation to exercise of New Zealanders during the first Covid-19 lockdown in 2020 and to obtain qualitative information on the reasons for those behaviours. You write a specific goal that it should be required a little more explanation.

Reply: As suggested by reviewer 1 we have clarified that the aim of the study was to describe the PA level and mental health status of adults during the first lockdown in New Zealand. Furthermore, we have clarified that we aimed to explore the reasons for the reported PA level and mental health status. We hope this adequately addresses your query.

  1. Materials and Methods

[Line 78-79] This was a cross-sectional study design in which an anonymous survey was distributed to adults over the age of 18 years in the general adult NZ population. You could mention from what age you consider adults and explain why have you chosen the general population?

Reply: In New Zealand those over the age of 18 are considered adults.

In Line 104-106: We have added the reason for choosing this population and age group. “This age group and population was chosen as no research has yet been undertaken regarding the impact of COVID-19 on health outcomes in New Zealand.”

2.1.Survey participants and distribution is a specific and good point. I would like only to mention some brief aspects to consider:

[Line 93-94] A modified form of the International Physical Activity Questionnaire (IPAQ) was used to collect information on domain-specific physical activity and sedentary behaviour over the last seven days during lockdown. It may be a good suggestion to write which are the aspects that have been modified or at least to mention the relevant ones.

Reply: Thank you for this suggestion. The only part of the form that was modified was the lead-in to the individual questions which asked to specifically refer the respondents’ answers to a seven-day period during lockdown level 3 or 4, which were the highest alert levels in New Zealand. This has been clarified in the manuscript (Line 122-124).

[Line 101] You could mention which are the six styles of motivation.

Reply: We have added the six styles of motivation in lines 131-133.

The six styles of motivation are amotivation, external regulation, introjected regulation, identified regulation, integrated regulation and intrinsic motivation. “

  1. Results

[Line 131-132] Of the 263 respondents who started the survey 248 completed (78% females) their responses and were included in the statistical analysis. Do we know the reasons why did they not answer the survey?

Reply: Unfortunately we do not have the reasons for not completing the survey and cannot infer about it in the manuscript.

  • Quality of life.

[Line 173-174] There were no differences in physical and mental health scores between male and 173 female respondents (p=0.33 and p= 0.46 respectively). I think the answers between female and male are not well represented because there is a huge difference responses. Therefore, I think it is important to consider the number of representation for each one.

Reply: We appreciate the reviewers comments and understand that this is an important consideration as more females than males responded to our survey. However, as suggested by reviewer 1, we have decided to not compare the different demographics as this was not a stated aim of the study. We have therefore removed this comparison.

  1. Discussion

[Line 261-262] Some of these barriers to physical activity were exemplified in the change in environment attributable to the COVID-19 lockdown. You could mention the differences between before and meanwhile the COVID-19.

Reply: The reviewer makes a fair comment, however we did not collect information that could extend our comments as suggested. However we have compared the reported levels of PA to lockdown to another similar study that had requested pre-to post covid levels of PA in New Zealand on lines 368-370.

[Line 305-306] This is in line with other research that has shown the benefits of physical activity on mental health [45, 46]. Here, you could add the articles that I mention to read in order to enrich some of them.

Reply: We thank the reviewer for the suggestion. We have included the Faulkner study as suggested by the reviewer. Lines 368-370.

Reviewer 2 Report

Thank you for the opportunity to read this very interesting paper.  The paper has many strengths that should be of interest to the journal audience. However, it is not a significant sample although it has a significance of content and the methods and results have been relevance. 

Author Response

(The authors gave the same response as above.)

Round 2

Reviewer 1 Report

Thank you for the opportunity to review the revised manuscript. I think the paper requires some additional work. Professional editing for language/more concise writing style may also be valuable. Specific points below:

  • Abstract: the aim is rather vague. What is meant by "health-related outcomes". Presumably you mean mental health?
  • Please clarify this sentence. "Of the 60% of respondents who reported exercise and physical activity as having a positive effect on their overall wellbeing during lockdown, approximately half (54%) reported that it positively benefitted their mental wellbeing, while 6% reported a positive benefit to body image or fitness." It reads as though half of the 60% (so 30% of total respondents) reported that PA benefitted their mental wellbeing, and 3.6% of total population reported a positive benefit to body image or fitness. Or is it simply that 54% and 6% of total respondents (thus adding to 60% in total who reported that physical activity had a positive mental health effect?? Suggestion: Fifty-four percent of respondents reported that physical activity positively benefitted their mental wellbeing, while 6% reported it benefitted their body image or fitness. Could a respondent select both options (i.e. that both mental health and fitness were benefitted or could they only select one main benefit)?
  • Line 19: can remove "compared to before lockdown".
  • Last two sentences - suggest merging and re-wording to: "While 60% of respondents reported that exercise and physical activity during lockdown had a positive effect on their overall wellbeing, more than half of respondents did not maintain their usual level of exercise or physical activity."
  • Given an aim was to "describe...reasons for those outcomes", can you provide a conclusion in Abstract regarding this?

Intro:

  • line 50 regarding surveys conducted in other countries - should the citations be from 10-26, not just 10-16?
  • line 51. What is meant by "specifically"?
  • line 52: ref 12 also reported a decline in self-reported physical activity early in the pandemic compared to before, but is not included in this citation series. 
  • line 56: remove comma. 
  • line 66-68. please re-word. presume you mean visits from close relatives or health care workers were now allowed (under alert level 3)?
  • line 68-69: was the lockdown completely lifted at this time (return to pre-pandemic lifestyles)? please clarify in paper. Also, "first lockdown" sounds like the first stage of this early lockdown (alert level 4). Presumably there was another lockdown later on and this is what you mean by the March-May period being the "first lockdown"?
  • line 71: suggest reword to "...and the underlying reasons."
  • line 73: move "motivation to exercise" to straight after "self-reported physical activity levels". This would make more sense logically - i.e. you investigated PA levels, motivation for PA, reasons for inactivity, and mental and physical wellbeing. In light of this, i suggest re-wording "quality of life" to "physical and mental health". Please be very specific with the aim throughout the paper - abstract, intro, and discussion (e.g. line 224-225 sounds as though you directly compared to pre-pandemic levels). Might help to word it exactly the same each time it is mentioned, as i still remain somewhat unclear with respect to the aim. 

Results:

  • line 135: add comma after "survey"
  • line 137: "the amount of self-137 reported physical activity participation". can this just say "physical activity levels". Also, abstract and introduction sometimes state "physical activity or exercise" but you report only on PA. Can you remove the word "exercise" throughout?
  • line 140: add comma after "arrangements"
  • line 142: full stop after "73% had children" New sentence re work, given this presumably has nothing to do with "living arrangements" which the prior sentence is about. Also % for working respondents is incorrect. It is 79% is the table and written as 72% in text. Also 0.8% missing here (79.1+20.1 = 99.2%)
  • Table 1: Round Asian % to one decimal place as per all others.
  • line 155: compared with females?
  • What are the QoL scores out of? 100? Suggest stating this in methods, results and Table 3 legend. 
  • Suggest present motivation to exercise data before QoL data. So it flows on from the self-reported PA levels. 
  • Table 4: sometimes 2 decimals presented, other times, just 1. Be consistent.
  • Line 165: what do you mean by a "shift"? Sounds like a shift compared to something else, like pre-pandemic life? Or do you just mean more people reported as being intrinsically motivated to undertake PA?
  • Line 176: space missing between regimen-and. Remove (202/238). Stating percentage only in text will suffice.
  • line 177: remove (n=202) and state "when these respondents were asked..."
  • section 3.5 in general is hard to read. Many % presented, etc. Can the authors consider revising this thoroughly to make it more concise, to the point, etc? 

Discussion

  • reconsider wording of aim. 
  • this line is not necessary: "The findings from this study indicated that the lockdown in New Zealand did result in a change in reported physical activity levels."
  • line 322: reword: "The management of the crisis in New Zealand likely impacted the effects on aspects of health."
  • Throughout Discussion, please revise with better clarity and conciseness. 

Author Response

Thank you for the opportunity to review the revised manuscript. I think the paper requires some additional work. Professional editing for language/more concise writing style may also be valuable.

Reply: We are grateful to the reviewer for the time invested in the peer review of our manuscript, and the constructive feedback provided. In incorporating the suggestions made, we have edited the manuscript language and writing style and describe the changes in our response to the specific comments below.

Specific points below:

  • Abstract: the aim is rather vague. What is meant by "health-related outcomes". Presumably you mean mental health?

Reply: Apologies for not being specific.

The abstract has been rewritten as follows:

“This study describes self-reported physical activity (PA), motivation to exercise, physical and mental health and feelings towards PA during the March-May 2020 COVID-19 lockdown in New Zealand. Adults over the age of 18 years (n=238; 80.2% female) completed the International Physical Activity Questionnaire (IPAQ), the Behavioural Regulation in Exercise Questionnaire 3, the Short Form-36 and open-ended questions about PA through an anonymous online survey. 85% of respondents took part in regular PA prior to lockdown, but only 49.8% were able to maintain their usual level of PA. Although respondents were considered sufficiently physically active from the IPAQ, 51.5% reported not being able to maintain their usual level of PA primarily due to the closure of their gym facilities. Sixty percent of respondents reported that PA had a positive effect on their overall wellbeing. When asked to specify which aspects of wellbeing were affected, the effect on mental health was reported the most while the effect on body image or fitness was reported the least. Strategies to increase or maintain engagement in physical activity in lockdowns should be encouraged to promote positive mental health during the COVID-19 pandemic.”

As suggested by the reviewer we have also kept the aim consistent throughout the manuscript in the introduction, results and discussion sections.

  • Please clarify this sentence. "Of the 60% of respondents who reported exercise and physical activity as having a positive effect on their overall wellbeing during lockdown, approximately half (54%) reported that it positively benefitted their mental wellbeing, while 6% reported a positive benefit to body image or fitness." It reads as though half of the 60% (so 30% of total respondents) reported that PA benefitted their mental wellbeing, and 3.6% of total population reported a positive benefit to body image or fitness. Or is it simply that 54% and 6% of total respondents (thus adding to 60% in total who reported that physical activity had a positive mental health effect?? Suggestion: Fifty-four percent of respondents reported that physical activity positively benefitted their mental wellbeing, while 6% reported it benefitted their body image or fitness. Could a respondent select both options (i.e. that both mental health and fitness were benefitted or could they only select one main benefit)?

Reply:  We apologise for failing to provide clear information. We have since amended the sentence as below.

Sixty percent of respondents reported that PA had a positive effect on their overall wellbeing. When asked to specify which aspects of wellbeing were affected, the effect on mental health was reported the most while the effect on body image or fitness was reported the least.

Respondents could only have one domain option assigned to their answer. We have clarified this in the methods section (line 232).

  • Line 19: can remove "compared to before lockdown".

Reply: We have amended the text accordingly.

  • Last two sentences - suggest merging and re-wording to: "While 60% of respondents reported that exercise and physical activity during lockdown had a positive effect on their overall wellbeing, more than half of respondents did not maintain their usual level of exercise or physical activity."

Reply: We thank the reviewer for the suggestion. We have concluded the abstract with the following statement, “Strategies to increase or maintain engagement in physical activity during lockdowns should be encouraged to promote positive mental health during the COVID-19 pandemic.

  • Given an aim was to "describe...reasons for those outcomes", can you provide a conclusion in Abstract regarding this?

Reply: We thank the reviewer for the suggestion. In re-writing the abstract, we have since concluded the abstract as follows: “Strategies to increase or maintain engagement in physical activity during lockdowns should be encouraged to promote positive mental health during the COVID-19 pandemic.”

Intro:

  • line 50 regarding surveys conducted in other countries - should the citations be from 10-26, not just 10-16?

Reply: We have rewritten lines 54-69 for conciseness and clarity and have amended this reference citation in line 60.

  • line 51. What is meant by "specifically"?

Reply: It is unnecessary to have the word specifically here and it has been removed. Line 62

  • line 52: ref 12 also reported a decline in self-reported physical activity early in the pandemic compared to before, but is not included in this citation series. 

Reply: Thank you for pointing this out, in response to the comment above, we have added ref 12 to the in-text references in line 63.

  • line 56: remove comma. 

Reply: The paragraph (lines 54-69) has been rewritten for conciseness.

  • line 66-68. please re-word. presume you mean visits from close relatives or health care workers were now allowed (under alert level 3)?

Reply: Thank you for pointing this out – we have changed lines 77-81 to state: “New Zealand moved to alert level 3 on 27 April 2020, which still imposed the same restrictions as alert level 4, except visits from close relatives or health care workers were allowed.

  • line 68-69: was the lockdown completely lifted at this time (return to pre-pandemic lifestyles)? please clarify in paper. Also, "first lockdown" sounds like the first stage of this early lockdown (alert level 4). Presumably there was another lockdown later on and this is what you mean by the March-May period being the "first lockdown"?

Reply: We have clarified on lines 80-81 that when the lockdown was lifted it was to pre-pandemic levels but with border closure.

In response to your second comment, the first lockdown refers to the first of 2 lockdowns (entry into Alert levels higher than 2) that New Zealand has so far entered into. We have removed the word first lockdown as it is irrelevant to this study. We have also removed the word “first” later on in the paragraph (lines 83-84 and 87) and specified that we are referring to the March -May lockdown period.

  • line 71: suggest reword to "...and the underlying reasons."

Reply: Thank you for this suggestion. We have re-worded line 82-85 to “The levels and underlying reasons for those levels of exercise or physical activity during the March-May COVID-19 lockdown of 2020 in New Zealand are unknown”.

We have then also reworded the last sentence of the introduction on lines 82-86 to “The aim of this study was to describe the self-reported physical activity levels, motivation to exercise and physical and mental health of New Zealanders and to obtain qualitative information on the feelings towards physical activity and exercise during lockdown"

  • line 73: move "motivation to exercise" to straight after "self-reported physical activity levels". This would make more sense logically - i.e. you investigated PA levels, motivation for PA, reasons for inactivity, and mental and physical wellbeing. In light of this, i suggest re-wording "quality of life" to "physical and mental health". Please be very specific with the aim throughout the paper - abstract, intro, and discussion (e.g. line 224-225 sounds as though you directly compared to pre-pandemic levels). Might help to word it exactly the same each time it is mentioned, as i still remain somewhat unclear with respect to the aim. 

Reply: Thank you for this important point. We have moved the motivation to exercise results to after the self-reported physical activity results and now appears on lines 175-179. We have also made adjustments throughout the manuscript.

As we have reworded the aim we have used physical and mental health in place of quality of life where relevant, while the qualitative comments are referred to in the aims as respondents’ feelings towards physical activity and exercise during lockdown.

Results:

  • line 135: add comma after "survey"

Reply: Comma added line 152.

  • line 137: "the amount of self-137 reported physical activity participation". can this just say "physical activity levels". Also, abstract and introduction sometimes state "physical activity or exercise" but you report only on PA. Can you remove the word "exercise" throughout?

Reply: We agree with this suggestion and have changed the sentence in lines 154-155. We have removed the word exercise from the results referring to physical activity measured by the IPAQ, however in some qualitative comments people reported on their feelings towards physical activity and/or exercise and so we have chosen to keep the mention of exercise when referring to the qualitative results.

  • line 140: add comma after "arrangements"

Reply: Comma added now in line 157.

  • line 142: full stop after "73% had children" New sentence re work, given this presumably has nothing to do with "living arrangements" which the prior sentence is about. Also % for working respondents is incorrect. It is 79% is the table and written as 72% in text. Also 0.8% missing here (79.1+20.1 = 99.2%)

Reply: Line 158-160, new sentence rewritten as: “79% of respondents were working during lockdown.”

Percentage of respondents who were working corrected has now been corrected to 79.1% on line 159.

  • Table 1: Round Asian % to one decimal place as per all others.

Reply: Apologies for this – now reads 7.1%

  • line 155: compared with females?

Reply: The reviewer is correct. We have added “…than females” at the end of the sentence, line 173.

  • What are the QoL scores out of? 100? Suggest stating this in methods, results and Table 3 legend. 

Reply: The QoL scores are out of 100 with 100 indicating the most favourable health state. We have stated the scores are out of 100 in the methods on lines 127-129 and in the results table 4 title and legend.

  • Suggest present motivation to exercise data before QoL data. So it flows on from the self-reported PA levels. 

Reply: Thank you for this suggestion and we have moved the motivation data to before QoL and adjusted the Table numbers.

  • Table 4: sometimes 2 decimals presented, other times, just 1. Be consistent.

Reply: Apologies for this inconsistency. All decimals in Table 4 (new Table 3) have been changed to 1 decimal place.

  • Line 165: what do you mean by a "shift"? Sounds like a shift compared to something else, like pre-pandemic life? Or do you just mean more people reported as being intrinsically motivated to undertake PA?

Reply: We have rewritten this sentence as “Overall, respondents reported higher scores (“very true for me”) in the questionnaire items that related to intrinsic motivation.” Lines 177-179

  • Line 176: space missing between regimen-and. Remove (202/238). Stating percentage only in text will suffice.

Reply: Space added in line 196. Numbers in parentheses have been deleted line 198.

  • line 177: remove (n=202) and state "when these respondents were asked..."

Reply: Lines 195-196.

  • section 3.5 in general is hard to read. Many % presented, etc. Can the authors consider revising this thoroughly to make it more concise, to the point, etc? 

Reply: In response to the suggestions made above about improving clarity throughout the document we have made changes to this section to improve its readability.

Discussion

  • reconsider wording of aim. 

Reply: The first two sentences on line 248-253 have been re-worded to: “The aim of the present study was to describe self-reported physical activity, motivation to exercise and physical and mental health of New Zealanders during the March-May 2020 COVID-19 lockdown. This study also aimed to explore some of the feelings towards physical activity during this lockdown period.

  • this line is not necessary: "The findings from this study indicated that the lockdown in New Zealand did result in a change in reported physical activity levels."

Reply: Deleted in line 252 – 253.

  • line 322: reword: "The management of the crisis in New Zealand likely impacted the effects on aspects of health."

Reply: We have rewritten this sentence as: “Overall, the movement restrictions put in place in New Zealand due to the global COVID-19 pandemic impacted the level of PA which may then have had effects on physical and mental health.” Lines 351-354

  • Throughout Discussion, please revise with better clarity and conciseness. 

Reply: We thank the reviewer for the suggestion and have revised the discussions accordingly.

Reviewer 2 Report

Thanks for correcting  all the suggestions and for taking them into consideration

Author Response

Thank you.